# Analysis of Morphological Traits, Mineralization, and Mechanical Properties of Femoral Bones in Young and Adult European Hares (*Lepus europaeus*)

**DOI:** 10.3390/ani13132077

**Published:** 2023-06-23

**Authors:** Cezary Osiak-Wicha, Ewa Tomaszewska, Siemowit Muszyński, Marian Flis, Marcin B. Arciszewski

**Affiliations:** 1Department of Animal Anatomy and Histology, Faculty of Veterinary Medicine, University of Life Sciences, Akademicka 12, 20-950 Lublin, Poland; mb.arciszewski@wp.pl; 2Department of Animal Physiology, Faculty of Veterinary Medicine, University of Life Sciences, Akademicka 12, 20-950 Lublin, Poland; ewarst@interia.pl; 3Department of Biophysics, Faculty of Environmental Biology, University of Life Sciences, Akademicka 13, 20-950 Lublin, Poland; siemowit.muszynski@up.lublin.pl; 4Department of Ethology and Wildlife Management, Faculty of Animal Sciences and Bioeconomy, University of Life Sciences, Akademicka 13, 20-950 Lublin, Poland; marian.flis@up.lublin.pl

**Keywords:** bone, bone mineralization, mechanical properties, locomotion, lagomorphs

## Abstract

**Simple Summary:**

To understand the developmental changes in hare femora, this study aimed to examine the mechanical properties, mineralization, and general geometric properties of the thigh bone (femur) throughout the growth and maturation process. The study involved analyzing the strength, stiffness, bone mineral density, and structure of the femur in both young and adult hares. Additionally, the study sought to establish a connection between bone properties and the ecological functions of hares in their natural habitats. Understanding the changes in hare femora as they grow and mature is crucial for many reasons as it provides valuable insights into the effects of aging on wild hares, shedding light on their adaptability and survival strategies.

**Abstract:**

Lagomorphs, which include hares, rabbits, and pikas, are herbivorous animals renowned for their exceptional running abilities. The femur, the largest and strongest bone in their bodies, plays a crucial role in supporting their weight and facilitating movement. This study aimed to investigate the structural and functional changes in the femora of hares during their development in a sex-dependent manner, and the influence of aging on femur structure and function. Various mechanical properties, including stiffness and strength, as well as densitometry and morphology, were evaluated. The study was conducted on *n* = 24 hares collected from a hunting district in the Lublin region of Poland and divided into four groups: young females, adult females, young males and adult males (*n* = 6 animals each). Dual-energy X-ray absorptiometry (DXA) was used to measure bone mineral content (BMC) and bone mineral density (BMD), and a three-point bending test was performed to assess mechanical properties. The findings revealed age-related differences in bone properties, with adult males exhibiting increased BMC, and BMD compared to young males. Geometrical properties of the femora mid-diaphysis, such as cortical index and cross-sectional area, remained relatively unchanged during maturation. Regarding mechanical properties, the femora of young males exhibited higher elastic work compared to those of young females, while the femora of adult males exhibited higher elastic and breaking work than those of adult females. The stiffness of femora was higher in adult females compared to young females. The results provide insights into the development and aging of hare femora and contribute to our understanding of the relationship between bone mechanical properties, musculoskeletal system, and aging in the wild. This knowledge can inform animal husbandry practices in captivity and enhance our broader understanding of the ecological functions of lagomorphs.

## 1. Introduction

Lagomorphs, like hares (e.g., *Lepus europaeus*), rabbits (e.g., *Oryctolagus cuiniculus*) or pikas (e.g., *Ochotona princeps*), are primary herbivorous animals that play critical roles as herbivores in the wild [1]. Their skeletal anatomy has been an area of interest in previous research, particularly for understanding how it relates to their performance and ecological functions [2,3,4,5]. Lagomorphs are known for their ability to reach high speeds, with hares being one of the fastest land mammals, capable of running up to speeds of 80 km/h [6,7]. This exceptional running ability is attributed to various adaptations in their musculoskeletal system, including long and powerful hindlimbs, elongated metatarsal bones and a unique arrangement of muscles that enable efficient propulsion, maneuverability and jumping as an evasive tactic [8,9,10]. This specialized adaptation for running is known as cursoriality [11]. 

The femur, being the largest and strongest bone in the body, plays a critical role in supporting body weight and facilitating movement. The femur of hares, in particular, is of interest due to its unique adaptations for rapid locomotion and jumping, characterized by a more structurally slender bone with decreased maximum bending strength [4]. It is likely that developmental changes in hare femora are part of peculiar adaptations of these animals for fast running and leaping, which may further change during the animal’s growth and maturation. 

The mechanical properties of bones are essential for their function in the animal body and encompass their composition, structure, and geometry, which determine their ability to resist forces and deformations [12,13]. Age is also a crucial factor that can significantly impact the mechanical properties of bones. As hares age, various physiological changes occur, affecting the morphology and structure of their bones [13]. However, the specific age- and sex-dependent variations in hare bone mechanical properties, morphology, and densitometry, especially in the femur, remain largely unexplored. 

The juvenile (young) phase of life poses significant challenges for numerous species. Once they become independent, immature animals must undertake essential survival tasks similar to those of adults, despite their smaller body size. Since juveniles have not yet reached reproductive maturity, we can expect the existence of strong selective pressures favoring mechanisms that compensate for these developmental limitations. These mechanisms enable individuals to overcome ontogenetic constraints and ultimately reach reproductive adulthood. 

In this study, our objective was to investigate the structural and functional changes in the femora of hares during their ontogeny, encompassing both males and females. We hypothesized that young hares possess specific mechanical properties in the bones of the hindlimb, particularly the femur, enabling them to achieve performance comparable to that of adult animals. To test this hypothesis, we evaluated various parameters, including mechanical properties such as morphology, densitometry, stiffness, strength and elasticity. 

The findings of this study will not only shed light on how maturation influences femur structure in hares, but also provide insights into the variations in bone mechanical properties.

## 2. Materials and Methods

### 2.1. Characterization of the Research Area

The research material for this study consisted of femora belonging to European brown hare (*Lepus europaeus europaeus* (Pallas, 1778)), the only hare species found in Poland. The femora were obtained through hunting in the western part of the Lublin region, which is known for having one of the highest concentrations of hares in Poland [14]. As a result, the animals are culled annually in this area. The hunting district was located near the town of Opole Lubelskie (51°08′51″ N, 21°58′08″ E). This region is characterized by a relatively low forest cover, not exceeding 25% [14]. Additionally, the landscape features numerous wastelands, small wooded enclaves, and forest complexes [15].

### 2.2. Research Material 

The research material consisted of deceased hares shot during the hunting season in December 2020, in accordance with the provisions of the hunting law in Poland (Act–Law hunting 1995 https://isap.sejm.gov.pl/isap.nsf/DocDetails.xsp?id=wdu19951470713; accessed on 22 June 2023, Regulation of the Minister of Environment 2005 https://isap.sejm.gov.pl/isap.nsf/DocDetails.xsp?id=WDU20050610548; accessed on 22 June 2023). Bone samples for testing were obtained from deceased animals, eliminating the need to obtain the consent of the Ethics Committee. 

Due to the COVID-19 restrictions implemented in 2020, the animals were shot by individual hunters in accordance with the decision of the Minister of Climate and Environment in 15 December 2020 (DLŁ-ZŁ.4142.1.2020). All hares were weighed using a digital scale (HCB20K10, Kern, Balingen, Germany) with an accuracy of 0.1 kg immediately after shooting. 

The sex and age of each individual animal was assessed. The sex was determined by observing secondary sex characteristics. Age was assessed using the method which relies on the presence or atrophy of the cartilaginous epiphysis on the ulnar bone of the forelimb, and was additionally verified using the lense weight method [16,17]. Youngs are those born in the same year as the hunting season; therefore, young hares are those born between March and August of the same year. Adult hares refer to those born in the previous year or earlier, as the typical survival period for individuals of this species does not exceed 3 years [18]. The first shot individuals that met the criteria for age and sex were used for the study (*n* = 24), ensuring random sampling.

After careful dissection, the femora from both legs were cleaned of any adhering tissues using scissors and scalpel, and then measured for length and weight. Bones were divided into four groups: young females, adult females, young males and adult males, each group contained *n* = 6 animals. Based on the measurements performed for the left femora, the relative bone weight (RBW), calculated as a bone weight to whole body weight ratio, and the Seedor index, calculated as a bone weight to bone length ratio as an indicator of whole bone density, were calculated. Next, the bones were individually wrapped with gauze soaked in 0.9% normal saline. These prepared bones were then placed in separate ziplock bags. The bones were stored in a freezer at −20 °C and reserved for bone densitometry analyses and mechanical testing at a later date. The left femur was designated for densitometric analysis, while the right femur was designated for 3-point bending test and measurements of mid-diaphysis geometry. Before the measurement, bones were thawed in a refrigerator at 4 °C overnight.

### 2.3. DXA Measurements

Bone mineral density (BMD) and bone mineral content (BMC) were measured using the dual-energy X-ray absorptiometry (DXA) method on a densitometer (Lunar iDXA, GE, Madison, WI, USA). Prior to the scanning procedure, the apparatus was calibrated with densitometer-specific phantoms (BMD range: 0.6–1.4 g/cm^2^) following the manufacturer’s instructions. The scanning was conducted in “small animal” mode using enCORE software (ver. 17.0; GE, Madison, WI, USA, https://services.gehealthcare.com/gehcstorefront/p/LU45563; accessed on 22 June 2023). The bone was placed on a special pad designed for “small animal” scans provided by the densitometer manufacturer during scanning. No additional objects were used to simulate the soft tissue surrounding the bone. The BMD and BMC measurements were obtained from the collected data using operator-defined regions of interest (ROIs) that covered the entire bone. All the measurements were performed by the same person.

### 2.4. Bone Analysis

To evaluate the mid-diaphysis bone mechanical properties, a 3-point bending test was conducted using a universal testing machine (Zwick Z010, Zwick, Ulm, Germany). The test was performed at a constant loading rate of 10 mm/min until the bone fractured, and the load-deflection curves were recorded to determine the mechanical properties of the femur: yield force, yield deflection, elastic work, and stiffness in the elastic region of deformation, as well as breaking force, breaking deflection, and breaking work at bone breakage [19]. After the test, the bones were cut at the midpoint of the bone diaphysis using a diamond bandsaw (MBS 240/E, Proxxon GmbH, Foehren, Germany) [20]. The external and internal transversal and anteroposterior diameters of the mid-diaphysis cross-section were measured using a digital caliper to determine mid-diaphysis geometric parameters: mean relative wall thickness (MRWT), cortical index (CI), cross-sectional area (CSA), and cross-sectional moment of inertia (Ix). Using the previously determined femur mechanical properties and calculated mid-diaphysis geometrical parameters, femur material properties were determined: yield strain, yield stress, breaking strain, and breaking stress [19].

### 2.5. Statistical Analysis

The mean values and standard errors (SE) were used to express all results. The normal distribution of the variables was assessed via the W Shapiro–Wilk test of normality; for normally distributed data, a one-way Analysis of Variance (ANOVA) with planned comparison (the so-called contrast analysis) was applied. The groups were compared by age and gender (young female–adult female, young male–adult male, young female-young male, adult female-adult male). Such a statistical approach allows for a reasonably reliable estimation of the significance of differences between the selected groups, while minimizing the increase in the *α* level. If the data were not normally distributed, the Kruskal–Wallis tests were performed. The overall differences were reported as significant at *p* < 0.05 and trends were noted when the *p*-values were <0.1. Statistica ver. 13.1 for Windows (TIBCO Software Inc., Palo Alto, CA, USA, https://www.statsoft.pl/statistica_13/; accessed on 22 June 2023) was used for all statistical analyses and a GraphPad Prism ver. 9.5.1 for Windows (GraphPad Software, San Diego, CA, USA, https://www.graphpad.com/features; accessed on 22 June 2023) was used to create the graphs.

## 3. Results

### 3.1. Bone Properties

Differences in body weight were only found between young and adult males (Figure 1A, *p* < 0.05). No significant differences were found between groups in terms of bone weight, bone length or Seedor index (Figure 1B,C,G). Adult males exhibited significantly lower RBW compared to young males (*p* < 0.01), while the difference between adult females was on the trend level (Figure 1D). Significant differences were also observed in BMD and BMC between young and adult males (*p* < 0.05), with higher levels of bone mineralization observed in older animals (Figure 1E,F, *p* < 0.05 for both). A trend toward significance was also observed for BMC of female femora, indicating greater bone mineralization in adult animals. 

### 3.2. Geometrical Properties

The only significant difference observed in the femur mid-diaphysis cross-sectional diameters was for the anteroposterior inner diameter between sexes in both age-matched groups, with a greater value of anteroposterior inner diameter of female femora (Figure 2D, *p* < 0.05 for both). A trend toward significance in the cross-sectional area of the femora was observed for young females and males (Figure 2E), with the femora of adult females showing greater values of CSA. Maturation from a young age to adulthood did not affect the mean relative wall thickness of the femur mid-diaphysis (Figure 2F), cortical index (Figure 2G) or moment of inertia (Figure 2H).

### 3.3. Mechanical Properties

There were no significant differences in yield force (Figure 3A) or breaking force (Figure 3D) between the groups. Significant sex-dependent differences were noted in elastic work, with the femora of males exhibiting higher elastic work than females, regardless of the animals’ age (Figure 3B, *p* < 0.001 and *p* < 0.01, for young and adult animals, respectively). There was also a significant difference in elastic work between the femora of males, with the femora of young individuals showing higher elastic work than the femora of adults. A significant difference in the stiffness of the femora was observed between young and adult females (Figure 3C, *p* < 0.05), with the femora of adult individuals showing higher stiffness. In males, this difference was observed at a trend level. Differences in femora breaking work were observed between adult females and adult males, and between males, which the femora of adult males showed lower values of breaking work (Figure 3E, *p* < 0.05 for both).

### 3.4. Bone Material Properties

The femora of hares from all groups showed similar values of yield strain (Figure 4A). Significant sex-dependent differences in yield stress were observed, with the femora of male individuals exhibiting greater values of yield stress compared to the femora of females, regardless of the animals’ age (Figure 4B, *p* < 0.05 and *p* < 0.01, for young and adult individuals, respectively). The femora of adult males exhibited lower breaking strain, compared to the femora of young males and adult females (Figure 4C, *p* < 0.01 for both). No differences in femora-breaking stress were observed between the groups (Figure 4D). At the trend level, the femora of adult males were characterized by greater values of Young’s modulus compared to young males (Figure 4E).

## 4. Discussion

Many studies have shown that mammals of all sizes exhibit similar stresses on their long bones during rapid locomotion. Most of these studies have focused on steady movement [21,22], although some have investigated hopping [23]. During the growth and maturation of an animal, its anatomical size and physiological parameters increase at different rates. Assuming that the bones of animals have geometric similarity, meaning their lengths increase in direct proportion to their diameters, it is generally accepted that peak stresses acting on them should increase as the size of the animal increases. This is because a bone’s strength, or its capacity to withstand compressive stress, is proportional to its cross-sectional area, while the forces acting on a bone are proportional to some multiple of body weight. Consequently, these forces increase at a faster pace than the bone’s ability to resist them [24].

Pikas, rabbits and jackrabbits exhibit notable morphological divergence, primarily associated with variations in the degree of cursoriality. Generally, pikas exhibit the lowest level of cursoriality, while jackrabbits and hares display the highest, and rabbits fall somewhere in between, occupying an intermediate position in terms of locomotor adaptation [25]. The diverse running abilities among lagomorphs arise from their distinct ecological niches. Pikas typically reside in burrows or rocky slopes (talus) that provide natural concealment. They rely on short, hopping locomotion between refuges to evade predators. Hares and jackrabbits, on the other hand, predominantly inhabit open environments such as grasslands and deserts. They rely on speed and endurance to escape predation. Rabbits, similar to pikas, swiftly navigate through woodland cover to obscure the view of predators, utilizing agility and quick movements [25,26,27]. These distinct locomotor adaptations reflect the specific strategies adopted by each lagomorph species in response to their respective habitats and predator pressures.

The mechanical properties of bones play a crucial role in locomotor performance, particularly in terms of force generation, energy storage, and transmission [28,29]. In the case of young hares, their hindlimb bones exhibit remarkable adaptations that contribute to their acceleration abilities, despite their smaller size. 

The survival of all young hares in the face of predatory threats is highly dependent on their ability to accelerate rapidly, enabling them to escape potential danger. Around 14–16 days of age, pups begin to develop their locomotive abilities and achieve juvenile independence by the third week [30]. However, during their first year of life, young hares face a significant predation threat similar to adult animals. Despite their smaller size and developmental constraints, young hares demonstrate impressive acceleration capabilities comparable to adult hares [27]. Their reduced body mass directly contributes to their superior performance while also mitigating the negative impact associated with increased body mass. The correlation between acceleration and power output related to mass has been observed in previous studies [28,31]. Additionally, the sturdy long bones of young hares enhance their ability to withstand bending loads.

Analyzed in this study bone samples were collected from specimens throughout the hunting season, which resulted in a range of ages for the young hares, approximately 5 to 9 months old. As a result, there may be significant variations in bone structure and development among young animals, despite belonging to the same age group. These developmental differences might have contributed to the lack of statistical significance in some of our findings. However, we did observe trends towards statistical significance, indicating potential differences between the juvenile and adult groups. We acknowledge that obtaining a more homogeneous group of younger hares would have strengthened the statistical significance and might further confirm our research hypothesis. Therefore, it is important to note that in some of the analyzed bone parameters, the difference between young and adult hares remained at the trend level.

In our study, we observed that the relative bone weight of the femora in young male hares is much higher than in their adult counterparts, while their length remains similar. As a result, young hares outperform their adult counterparts by generating a greater amount of mechanical power relative to their body mass. Furthermore, it is important to note that the lack of statistical significance in breaking force observed in this study confirms the research hypothesis, which assumed that there are no differences in mechanical strength of the femora of young and adult hares, despite lower bone mineral content or differences in bone size. It should be added that examined bone samples were collected from wild hunted animals, and there were restrictions on the age of the animals, such as a ban on hunting too young animals. This, along with the previously mentioned wide age range encompassing young individuals, could have contributed to the lack of a significant difference in bone length and weight between individuals of different ages.

One key finding of this study is the similar mechanical advantage of limb bones observed in immature hares compared to adult conspecifics. The mechanical advantage of a limb bone refers to its ability to produce a greater force output relative to the force applied to it [32,33]. In immature mammals, the limb bones exhibit a higher mechanical advantage, allowing for increased force production during locomotion. This enhanced muscle performance is likely linked to the specific demands of the immature stage, where individuals must navigate their environment and perform essential tasks despite their smaller size and limited physical capabilities. 

This study also demonstrated higher bone stiffness of the femora of adult females, which was not accompanied with lower values of BMD, indicating lower flexibility of the bones in young females. For male individuals, the difference in bone stiffness was observed at the trend level, despite differences in femora BMC and BMD between young and adult hares. These differences likely result from the observed variations in the geometric dimensions of the femora mid-diaphysis anteroposterior inner diameter between females and males, as the shape of the bone has a greater influence on its stiffness than the degree of mineralization [34]. The strength and elastic properties of bones are also critical determinants of acceleration capabilities. This study demonstrates, lower bone elasticity (in terms of yield stress), as well as lower elastic work (energy absorbed by bone during bending) in female individuals, irrespective of their age. All of the above indicates a higher susceptibility of female femora to bending. 

The structural properties of bone material, such as Young’s modulus, which describe the range of reversible plastic deformations of the bone, were significantly higher in the femora of adult males. This may allow male hares to withstand the higher forces generated during rapid acceleration without compromising the structural integrity of their bones. During the acceleration phase, the bones undergo cyclic loading, storing energy as they deform elastically. This stored energy is subsequently released during the propulsion phase, contributing to the forceful extension of the hindlimbs and resulting in rapid acceleration [34]. The efficient utilization of elastic energy reduces the metabolic cost of acceleration, allowing young hares to sustain high-speed bursts for extended periods. These bones efficiently transmit the forces generated by muscle contractions during each stride, enabling powerful propulsion and swift acceleration [34,35]. In conclusion, all above discussed differences in the values of bone elastic parameters between the bones of male and female individuals, both young and mature ones, require further research.

Hares are well-known for their unique method of locomotion, jumping, which contributes to their ability to move at high speeds and evade predators [10,36]. The morphology and mechanical properties of their bones play a critical role in this ability, particularly in their hindlimbs. The hindlimbs bones of hares are elongated and slender, with a significant proportion of their length occupied by the tibia and fibula. This elongation provides a long lever arm for force generation, allowing hares to generate substantial force to propel themselves forward during a jump [32,36]. Similar conclusions about the mechanical properties of hindlimb bones have been drawn from studies on other non-mammal animals, that exhibit jumping locomotion [37]. These findings highlight the importance of bone morphology and mechanical properties in facilitating the unique locomotor capabilities of hares during jumping.

Further research could explore the developmental changes in bone morphology and composition, as well as the interplay between muscle architecture and bone properties. These future studies should consider a broader range of young individuals by expanding the sample size and dividing them into more age-specific subgroups. This approach would contribute to a comprehensive understanding of the underlying mechanisms that drive the rapid growth and development observed in young hares.

## 5. Conclusions

Overall, the unique morphology and mechanical properties of hares’ femora, along with other adaptations, make them highly specialized for their method of movement. Despite the lower mineralization of their bones, immature hares exhibit distinct morphological and biomechanical characteristics that result in mechanical properties of their limb-long bones often comparable to those of adult conspecifics. These adaptations allow them to efficiently generate and transmit forces, store and release elastic energy and, ultimately, achieve rapid acceleration. The ability to evade predators during the vulnerable juvenile stage significantly enhances their survival. 

## Figures and Tables

**Figure 1 animals-13-02077-f001:**
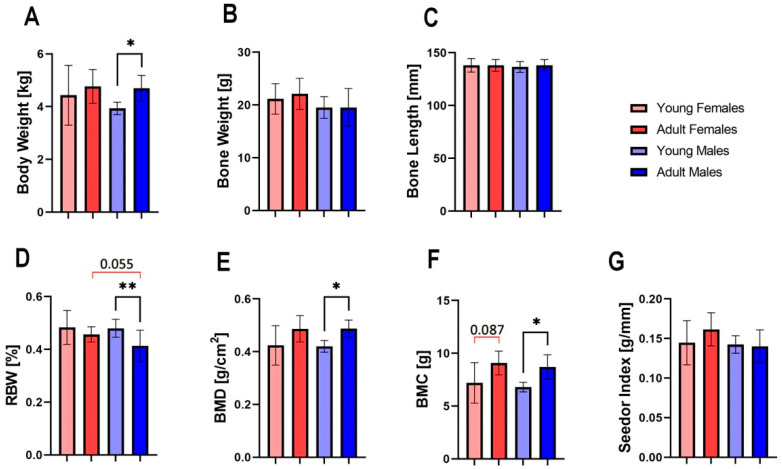
Body weight, general bone properties and mineralization of the femur of young female, adult female, young male, and adult male hares: (**A**) Body weight, (**B**) bone weight, (**C**) bone length, (**D**) relative bone weight (RBW), (**E**) bone mineral density (BMD), (**F**) bone mineral content (BMC), (**G**) the Seedor index. Asterisks (*) indicate significant differences between young and old animals of the same sex (* *p* < 0.05 ** *p* < 0.01), while red brackets indicate a trend towards significance between different groups (*p*-value between 0.05 and 0.1), where number above bracket indicates exact *p*-value.

**Figure 2 animals-13-02077-f002:**
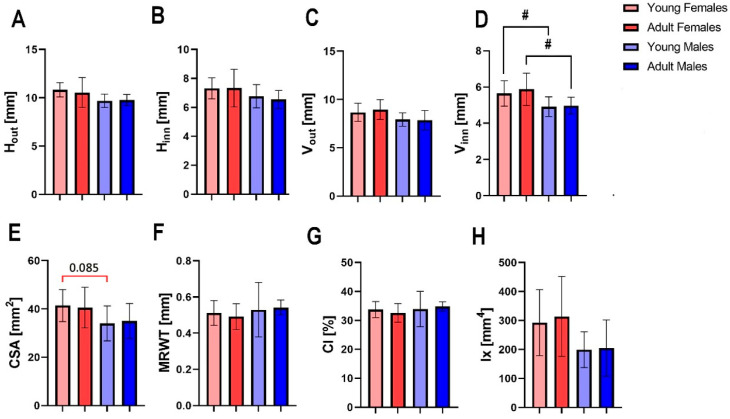
Geometrical properties of bone mid-diaphysis of the femur of young female, adult female, young male and adult male hares: (**A**) transversal outer diameter (H_out_), (**B**) transversal inner diameter (H_in_), (**C**) anteroposterior outer diameter (V_out_), (**D**) anteroposterior inner diameter (V_in_), (**E**) mid-diaphysis cross-sectional area (CSA), (**F**) mean relative wall thickness (MRWT), (**G**) cortical index (CI), (**H**) cross-sectional moment of inertia (Ix). Hashtags (#) indicate differences between specimens of different sexes at the same age (# *p* < 0.05), while red bracket indicates a trend towards significance between different groups (*p*-value between 0.05 and 0.1), where number above bracket indicates exact *p*-value.

**Figure 3 animals-13-02077-f003:**
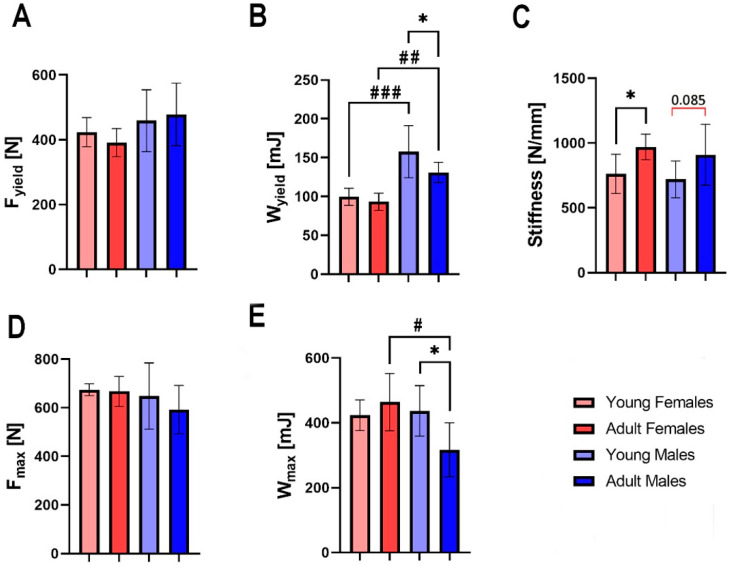
Mechanical properties of bone mid-diaphysis of the femur of young female, adult female, young male and adult male hares: (**A**) yield force (F_yield_), (**B**) elastic work (W_yield_), (**C**) stiffness, (**D**) breaking force (F_max_), (**E**) breaking work (W_max_). Asterisks (*) indicate significant differences between young and old animals of the same sex (* *p* < 0.05), hashtags (#) indicate differences between specimens of different sexes at the same age (# *p* < 0.05; ## *p* < 0.01; ### *p* < 0.001), and red bracket indicate a trend towards significance between different groups (*p*-value between 0.05 and 0.1), where number above bracket indicates exact *p*-value.

**Figure 4 animals-13-02077-f004:**
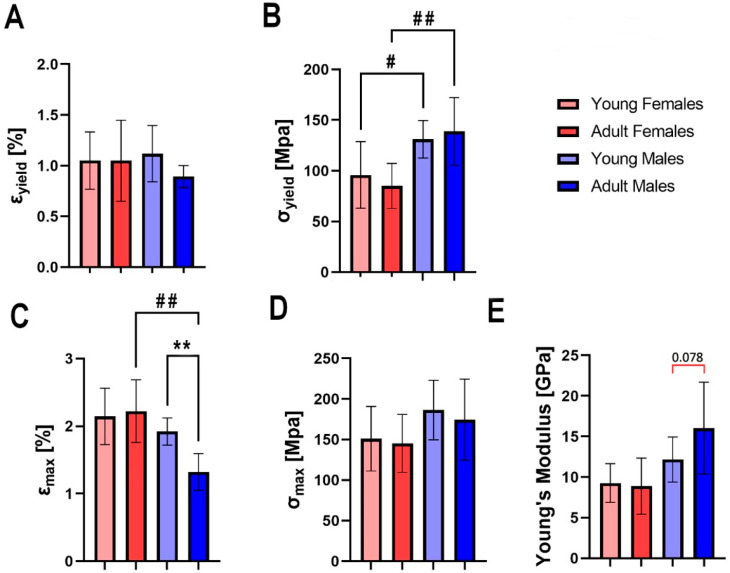
Bone material properties of the femur of young female, adult female, young male, and adult male hares: (**A**) yield strain (ε_yield_), (**B**) yield stress (σ_yield_), (**C**) breaking strain (ε_max_), (**D**) breaking stress (σ_max_), (**E**) Young’s modulus. Asterisks (*) indicate significant differences between young and old animals of the same sex (** *p* < 0.01), hashtags (#) indicate differences between specimens of different sexes at the same age (# *p* < 0.05; ## *p* < 0.01), and red bracket indicate a trend towards significance between different groups (*p*-value between 0.05 and 0.1), where number above bracket indicates exact *p*-value.

## Data Availability

Not applicable.

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
