# Peer review of "Analysis of Morphological Traits, Mineralization, and Mechanical Properties of Femoral Bones in Young and Adult European Hares (*Lepus europaeus*)"

_animals, 2023, doi:10.3390/ani13132077_

Round 1
Reviewer 1 Report
The authors of the submitted manuscript address an interesting topic concerning the mechanical strength of femoral bones in young and adult European hares. The research methods used in the study are appropriate, and the obtained results are generally well presented and discussed. I have only a few minor comments regarding the presentation and discussion of the trend. It is stated that the trend was introduced to highlight the remaining, albeit slight, differences between young and adult individuals, where these differences still fall slightly below the level of significance. Overall, this approach is proper and correct, taking into account the mentioned limitations in obtaining material from young animals due to legal protection of the wild animal species. However, the descriptions in the results and discussion regarding the detection of trends could be improved. For example, in the results, for L170: "there were no significant differences in bone length," and in L178: "the trend shows that bone length is similar between animals, despite different age, both in male and female hares." These result descriptions (and others) should be revised. For the mentioned example, a clearer description could be: "The difference in bone length between old and young individuals of both sexes was significant at the trend level." Such a description would be more understandable for readers. Similarly, in the discussion, for example, L299 could be improved to: "..while bone lengths were even shorter, albeit only at the trend level." Additionally, if possible, the existence of the trend should be mentioned more frequently in the discussion. Other comments: L60 gracile – consider using different word L95 “E).” L117-18 “tissues and measured for length…” L118 “four groups: young females ….” - remove bracket Left and right bones were collected. Were all of them used for dendrometry and mechanical testing, or were they divided for both tests? L142 “constant loading rate” L145, 150, 153: reference(s) where appropriate equations are presented is(are) needed L152 “determined: yield…” - remove bracket L303 In the contrary, the lack of statistical significance confirms the hypothesis stated at the beginning of the study, which suggests that there are no differences in mechanical strength despite differences in bone size. Please correct this, as it is a very important observation. Please also revise other fragments of the discussion. L309 Remove this sentence or correct to : “is the similar mechanical properties” L317-318 Remove this sentence
Author Response
Dear Reviewer,
We appreciate the time and effort that you have dedicated to providing your valuable feedback on our manuscript. We are grateful for your insightful comments on our paper. We have been able to incorporate changes to reflect most of the suggestions provided. We have highlighted in red the changes within the manuscript.
Here is a point-by-point response to Reviewer comments and concerns.
The authors of the submitted manuscript address an interesting topic concerning the mechanical strength of femoral bones in young and adult European hares. The research methods used in the study are appropriate, and the obtained results are generally well presented and discussed. I have only a few minor comments regarding the presentation and discussion of the trend. It is stated that the trend was introduced to highlight the remaining, albeit slight, differences between young and adult individuals, where these differences still fall slightly below the level of significance. Overall, this approach is proper and correct, taking into account the mentioned limitations in obtaining material from young animals due to legal protection of the wild animal species.
We thank the Reviewer for their positive feedback.
However, the descriptions in the results and discussion regarding the detection of trends could be improved. For example, in the results, for L170: "there were no significant differences in bone length," and in L178: "the trend shows that bone length is similar between animals, despite different age, both in male and female hares." These result descriptions (and others) should be revised. For the mentioned example, a clearer description could be: "The difference in bone length between old and young individuals of both sexes was significant at the trend level." Such a description would be more understandable for readers.
We would like to express our gratitude to the Reviewer for their highly valuable comment. As a result of new statistical analyses, the entire results section has undergone significant revisions, and as such, this and similar descriptions of the results have been corrected and rephrased.
Similarly, in the discussion, for example, L299 could be improved to: "..while bone lengths were even shorter, albeit only at the trend level." Additionally, if possible, the existence of the trend should be mentioned more frequently in the discussion.
Other comments:
L60 gracile – consider using different word
Corrected to “slender”.
L95 “E).” L117-18 “tissues and measured for length…”
Corrected.
L118 “four groups: young females ….” - remove bracket Left and right bones were collected. Were all of them used for dendrometry and mechanical testing, or were they divided for both tests?
The standard practice in our laboratory involves using one bone for densitometric measurements and another bone for mechanical tests, as these measurements are often conducted at different times and require different preparation methods for the bones. Similarly, in the present study, the left femora were used for densitometry, while the right femora were subjected to three-point bending tests and measurements of mid-diaphyseal geometry. We apologize for not providing these key details earlier, and they have now been corrected.
L142 “constant loading rate” L145, 150, 153: reference(s) where appropriate equations are presented is(are) needed
The corresponding reference has been included in the corrected version.
L152 “determined: yield…” - remove bracket
Corrected.
L303 In the contrary, the lack of statistical significance confirms the hypothesis stated at the beginning of the study, which suggests that there are no differences in mechanical strength despite differences in bone size. Please correct this, as it is a very important observation. Please also revise other fragments of the discussion.
We thank the Reviewer for their highly valuable comment. Due to new statistical analyses, the whole discussion has been significantly revised and this and other similar errors have been corrected.
L309 Remove this sentence or correct to : “is the similar mechanical properties”
This sentence has been rephrased.
L317-318 Remove this sentence
This sentence has been removed.
Reviewer 2 Report
The manuscript entitled “Analysis of Morphological Traits, Mineralization, and Mechanical Properties of Femoral Bones in Young and Adult European Hares (Lepus europaeus)" has the main objective of testing the morphological traits, mineral composition and mechanical properties of femora among groups of males and females of youngsters and adults of Lepus europaeus. Authors related these traits with the ecology and biology of the group, particularly with the pre- and postmaturation size.
The topic of the present manuscript is attractive and ambitious. In addition, the methodology used is the most suitable for the established purpose. I like how the manuscript is written and organized. The introduction provides the most important ideas in the field, what is wonderful for future non-specialized readers. The discussion has a clear guiding thread, and I specially thank all the figures that authors have included for a clear understanding of the topic. The authors reach the main goal of the research successfully, and I would like to congratulate them for this interesting study.
The only problem that I have detected, and that, from my point of view, the authors can solve in a future reviewed version of the paper is about the statistic tests that they have carried out. I have understood that the authors only used two-tailed Students T-test (or non-parametric version) though they have more than 2 groups of analysis: young females, young males, adult females and adult males. Statistically, they should test the differences among groups using an ANOVA (or non-parametric version in the case of non-normal distributions). This is because when performing multiple T-tests in the same data, the Type I error (probability of rejecting the null hypothesis given that it is true) increases. If you carry out a simple T-test this error is of 5%, but when you perform a second T-test with the same data this increases to 10%, and so on. In the case of the presented manuscript, they carried out 4 T-test with the same data: between young females and young males; between young males and adult males; between young females and adult females; and between adult males and adult females. So, the Type I error raised to 20%. Particularly, ANOVA has the advantage to maintain the Type I error at 5% and you can conduct multiple comparison among groups (posthoc analysis). For this reason, I suggest that authors change this point of the methodology, and use ANOVA to obtain results that are more reliable.
Besides, I have added some suggestion that I would like that authors will take in consideration, because I think that them would improve the quality of the paper:
- Detail the taxonomical attribution of the specimens to Lepus europaeus.
- Detail the concept mechanical advantage. It is directly in the discussion, but which variables that you measure point to a greater mechanical advantage?
- Correct anatomical concepts: vertical diameter = anteroposterior diameter and horizontal diameter = transversal diameter.
- In figures, one of them lacks the numbers in the X-axis, or in others you are talking about a section of a Figure that does not exist.
I am attaching the line-by-line review (pdf) with these suggestions and others, in addition to details, questions, grammatical errors, etc.
Hope to see this ms published soon,

I think that the English is good.
Author Response
Dear Reviewer,
We appreciate the time and effort that you have dedicated to providing your valuable feedback on our manuscript. We are grateful for your insightful comments on our paper. We have been able to incorporate changes to reflect most of the suggestions provided. We have highlighted in blue the changes within the manuscript.
Here is a point-by-point response to Reviewer comments and concerns.
The manuscript entitled “Analysis of Morphological Traits, Mineralization, and Mechanical Properties of Femoral Bones in Young and Adult European Hares (Lepus europaeus)" has the main objective of testing the morphological traits, mineral composition and mechanical properties of femora among groups of males and females of youngsters and adults of Lepus europaeus. Authors related these traits with the ecology and biology of the group, particularly with the pre- and postmaturation size.
The topic of the present manuscript is attractive and ambitious. In addition, the methodology used is the most suitable for the established purpose. I like how the manuscript is written and organized. The introduction provides the most important ideas in the field, what is wonderful for future non-specialized readers. The discussion has a clear guiding thread, and I specially thank all the figures that authors have included for a clear understanding of the topic. The authors reach the main goal of the research successfully, and I would like to congratulate them for this interesting study.
We thank the Reviewer for their positive feedback.
The only problem that I have detected, and that, from my point of view, the authors can solve in a future reviewed version of the paper is about the statistic tests that they have carried out. I have understood that the authors only used two-tailed Students T-test (or non-parametric version) though they have more than 2 groups of analysis: young females, young males, adult females and adult males. Statistically, they should test the differences among groups using an ANOVA (or non-parametric version in the case of non-normal distributions). This is because when performing multiple T-tests in the same data, the Type I error (probability of rejecting the null hypothesis given that it is true) increases. If you carry out a simple T-test this error is of 5%, but when you perform a second T-test with the same data this increases to 10%, and so on. In the case of the presented manuscript, they carried out 4 T-test with the same data: between young females and young males; between young males and adult males; between young females and adult females; and between adult males and adult females. So, the Type I error raised to 20%. Particularly, ANOVA has the advantage to maintain the Type I error at 5% and you can conduct multiple comparison among groups (posthoc analysis). For this reason, I suggest that authors change this point of the methodology, and use ANOVA to obtain results that are more reliable.
We would like to express our sincere appreciation to the Reviewer for their highly valuable comment. In each of our conducted experiments, we carefully select the statistical model that best describes our data. However, we concur with the Reviewer that in this particular case, we may not have chosen the optimal statistical model for analyzing our data.
As recommended, we have now employed ANOVA to obtain more reliable results. To ensure appropriate analysis, we performed ANOVA with planned comparisons using appropriate contrast coefficients. We specifically focused on pre-planned selected comparisons instead of conducting multiple t-tests with Bonferroni correction as a post-hoc test. It is important to note that not all possible comparisons between groups were necessary. In total, we conducted four independent comparisons between two groups (YM vs AM, YF vs AF, YM vs YF, AM vs AF), excluding, as previously, the comparisons of YM vs AF and YF vs AM.
We have now included the new results and provided a comprehensive discussion on these findings. We hope that the implementation of our new statistical model will meet the approval of the Reviewer.
Besides, I have added some suggestion that I would like that authors will take in consideration, because I think that them would improve the quality of the paper:
- Detail the taxonomical attribution of the specimens to Lepus europaeus.
We thank the Reviewer for this comment, it has been detailed in the revised version.
- Detail the concept mechanical advantage. It is directly in the discussion, but which variables that you measure point to a greater mechanical advantage?
The phrase "mechanical advantage" has been removed from the manuscript. We have included a more precise and detailed description of the observed differences in selected bone biomechanical traits in the revised version of the discussion.
- Correct anatomical concepts: vertical diameter = anteroposterior diameter and horizontal diameter = transversal diameter.
Thank you for pointing out this oversight, the description of concepts (planes) have been corrected.
- In figures, one of them lacks the numbers in the X-axis, or in others you are talking about a section of a Figure that does not exist.
We thank the Reviewer for pointing out these oversights. The chart for Young's modulus was moved to the next figure, but the description of the results was not removed after copying. It has been corrected. The X-axis for body weight chart has been also corrected.
I am attaching the line-by-line review (pdf) with these suggestions and others, in addition to details, questions, grammatical errors, etc.
We would like to extend our sincere gratitude to the Reviewer for their highly valuable comments. We have thoroughly reviewed the comments provided and have made the necessary revisions, addressing any ambiguities, grammatical errors, and other issues. In response to the feedback received, we have enhanced specific paragraphs in the discussion section and added a new paragraph at the end, highlighting potential avenues for future research.
Round 2
Reviewer 2 Report
The authors of the manuscript entitled “Analysis of Morphological Traits, Mineralization, and Mechanical Properties of Femoral Bones in Young and Adult European Hares (Lepus europaeus)” have submitted a new version, taking into account most of the suggested comments and making, when it was necessary under their point of view, the appropriate changes.
In this new version, I have identified two minor errors:
1) lin. 191-193: it is indicated that there is statistical difference between adult female and male, but in the figure 2E the difference is between young male and female.
2) Plese, review the section bibliography because there are names of species that are not in italics.
I would like to congratulate them for this interesting study. Hope to see it published soon.
Author Response
1) lin. 191-193: it is indicated that there is statistical difference between adult female and male, but in the figure 2E the difference is between young male and female.
Thank you for noticing that obvious error. It has been corrected to "young".
2) Plese, review the section bibliography because there are names of species that are not in italics.
Thank you. Corrected accoring Reviewer' suggestion.
